# Profiles of People Who Carried Out Late Primary Vaccination against COVID-19 in the Region of Murcia

**DOI:** 10.3390/vaccines11040732

**Published:** 2023-03-25

**Authors:** Amaya Bernal-Alonso, María Cruz Gómez-Moreno, Matilde Zornoza-Moreno, María Belén Laorden-Ochando, Francisca Isabel Tornel-Miñarro, Jaime Jesús Pérez-Martín

**Affiliations:** 1Instituto de Salud Carlos III, 28029 Madrid, Spain; 2Prevention and Health Protection Service, Regional Ministry of Health, 30008 Murcia, Spain

**Keywords:** vaccine hesitancy, COVID-19, COVID-19 vaccine

## Abstract

Despite the impact of the COVID-19 vaccination, vaccine hesitancy is a matter of concern. Despite a lower disease incidence, people continue to start primo-vaccination late. The aim of this study is to characterize people late primo-vaccinated and the reasons that led them to start vaccination. A quantitative, descriptive and prospective study was performed on the basis of phone surveys of people vaccinated from February to May 2022 in the Region of Murcia (Spain). The survey included socio-demographic and COVID-19 information, self-perception risk, vaccine security, Fear of COVID-19 Scale, reasons for not being vaccinated and reasons that have led them to vaccination. From a total of 1768 people receiving primo-vaccination, 798 people were contacted, and 338 people completed the survey. Among the interviewed people, 57% reported non-health-related reasons to get vaccinated, travel reasons being the primary one. The most reported health-related reason was a fear of COVID-19. There was a significant positive association between vaccination for health-related reasons and female gender (β = 0.72), cohabiting with a vulnerable person (β = 0.97), higher self-perceived risk (β = 0.13) and vaccine security dimension (β = 0.14). We identified two different profiles of people with late COVID-19 primo-vaccination, with health-related or non-health-related reasons. This work can be useful in designing specific communication strategies.

## 1. Introduction

Since the COVID-19 pandemic started, vaccination has proven to be the most effective way to lower the incidence of severe infections and deaths. It is estimated that vaccines, during 2021, saved 19.8 million lives all over the world and 456,200 lives in Spain [1]. Therefore, vaccination has become the main strategy in most countries for protecting the population from SARS-CoV-2 [2]. In Spain, the COVID-19 vaccination campaign started on 27 December 2020, when residents in nursing homes received their first shot. Since that day, the population has been progressively vaccinated according to the National COVID-19 Vaccination Strategy developed by the Spanish NITAG and approved by the Public Health Commission. This strategy was unique for the entire country. Since its first update, this strategy established priorities based on vulnerability and degree of exposure and, after that, on age group, both when the number of doses was limited and when there was greater availability. The strategy did not contemplate the possibility of citizens being able to choose the type of vaccine they receive, and they were given the vaccine that was available for their age group in the vaccination centre of their area [3]. Nevertheless, in 2022, the only vaccines administered in the Region of Murcia were mRNA ones. Public health vaccination policies in Spain do not utilize mandatory vaccination [4,5], and vaccine acceptance is very good, with coverage rates of over 95% for primary vaccination of vaccine-preventable diseases in childhood at the beginning of the pandemic [6]. Similarly, since the first update of the National COVID-19 Vaccination Strategy, COVID-19 vaccination in Spain, without prejudice to the duty of collaboration, was established as voluntary [7]. Vaccine hesitancy, defined as a behaviour influenced by a number of factors, including a lack of trust in the vaccine or the provider, complacency (not perceiving a need for the vaccine or not valuing it) or lack of convenient access [8], has been one of the main obstacles to this strategy, becoming a matter of concern for public health authorities, not only in Spain but all around the world. The figures for vaccine hesitancy in high-income countries vary between studies, but they can reach values higher than 40% in countries such as the USA, Italy and France [9,10]. In a COVID-19 vaccine hesitancy survey performed in 2021 in 23 countries with nationally representative samples of 1000 people for each one, vaccine hesitancy across all countries was associated with a lack of trust in COVID-19 vaccine safety and science and with a scepticism about its efficacy [11]. In a review of 15 studies [10], the most widely given reasons for vaccine hesitancy were being against vaccines in general, concerns about safety related to the fast development of COVID-19 vaccines, underestimation of the potential harm of SARS-CoV-2 infection, general lack of trust, doubts about the effectiveness of the vaccine, a belief that they are already immunized and doubt about the provenience of vaccines [12]. In July 2020, before the COVID-19 vaccination campaign began, according to a national survey, the Spanish population was favourable to a future vaccine against COVID-19: 32% of those surveyed were totally in favour of receiving it; 36% were in a favourable position, although with some reluctance; and 23% showed a high level of reservation. The remaining 9% preferred not to answer [13]. As the vaccination campaign developed, the level of acceptance of the vaccine among the Spanish population increased and, accordingly, COVID-19 vaccine coverage rates have been among the highest in the world [14]. In February 2022, 92.9% of people 12 years old or older had received at least one shot, and 92.6% had completed primary vaccination. Similarly, in the Region of Murcia, 93.2% of people 12 years old and older had received at least one shot, and 90.1% had completed primary vaccination [15].

In February 2022, when the study began, Spain was facing the end of the sixth pandemic wave, with indicators showing low rates of cumulative hospitalization, admissions in intensive care units (ICU) and mortality [16]. According to the COVID-19 weekly epidemiological report, on 26 January 2022, the Region of Murcia had reported 267,630 total COVID-19 cases and 1951 related deaths [17]. With these figures, Murcia was one of the autonomous communities least affected by COVID-19 in Spain. In addition, the month of February is usually, in the Region of Murcia, the last month of the flu vaccination campaign for people aged 60 years or older and those with risk conditions, with a low number of daily doses administered. Given the availability of doses, those people not belonging to risk groups who so wish can be vaccinated during this time. The end of this vaccination campaign coincided with the start of our study.

On 1 July 2021, the European Union implemented the EU Digital COVID-19 Certificate (obtained through vaccination, testing or recovery) in order to facilitate free circulation during the COVID-19 pandemic [18]. This COVID-19 Certificate, since its implementation, has been requested in many countries to enable people to travel. In the Region of Murcia, as in the rest of Spain, the application of the certificate was implemented for different activities, such as access to bars or restaurants. Nevertheless, the Region of Murcia had this certificate in place for less time and required it for fewer activities, not requesting it for access to different entertainment venues or for modification to capacity restrictions [19]. As the COVID-19 vaccination process advanced, this certificate was no longer required, and subsequently, in February 2022, the mandate of the use of masks in open areas was also withdrawn [20].

However, despite the decrease in the incidence of the COVID-19, severe cases and the restrictions associated with the pandemic, at the time the study began, in February 2022, there were still people who were just starting primo-vaccination. 

Vaccine hesitancy has been widely researched, but the reasons that lead to people to change their mind and accept vaccination have been less explored. The aim of this study is to characterize people beginning late primo-vaccination, as well as to determine the reasons that led these people to start primary vaccination this late into the pandemic, by which time vaccination was not as stringently required.

## 2. Materials and Methods

### 2.1. Study Design and Duration

A quantitative, descriptive and prospective study was performed based on phone surveys administered to people who received a late primo-vaccination from February to May 2022 in the Region of Murcia (Spain). Eligible people were those who started primo-vaccination between 7 February and 3 May 2022. These people’s data were extracted from the Region of Murcia Vaccination Registry’s Information System, called VACUSAN. The surveys were conducted by health professionals from the Vaccination Program. Two calls were made to each eligible person during the study period between 8 am and 3 pm from Monday to Friday.

The study was carried out with the authorization of all participants, and they were informed that the action of filling in the telephone questionnaire would be considered tacit informed consent for inclusion in the study. They were informed of the intention to publish the results and were given the opportunity to remove their questionnaire. The ethical aspects of the research were in accordance with the provisions of the Declaration of Helsinki of the World Medical Association on the ethical principles for medical research involving human beings and its subsequent amendments. Data were treated confidentially, in accordance with Spanish law. 

### 2.2. Population, Inclusion and Exclusion Criteria and Sample Size

An attempt was made to contact all 1768 people that began primo-vaccination in the region during the study period. After the two calls had been made to each person, 798 were successfully contacted. Inclusion criteria were having received primo-vaccination against COVID-19 in the selected period. Exclusion criteria were being disabled or under 18 and not speaking Spanish. Finally, we carried out the survey on 338 people.

### 2.3. Data Collection Tool

A cross-sectional phone-based study design was employed to gather data about hesitancy factors and attitudes towards COVID-19 vaccination in people with a late primo-vaccination in the Region of Murcia (Spain). Data were collected using an adapted and modified questionnaire based on literature and expert opinions. For the full questionnaire, a pilot study was performed in the general population, and questions with comprehension problems were redrafted. The data collection sheet consisted of seven parts (Appendix A). The first part included socio-demographic information. The second set of questions focused on participants’ COVID-19 variables, the third one focused on self-perception risk, the fourth one addressed vaccine security and the fifth one assessed the Fear of COVID-19 Scale (FCV 19-S) [21]. Finally, the sixth and the seventh sets attempted to gauge reasons for not being vaccinated earlier and reasons that have led the people to accept recent vaccination, respectively. For this last set of questions, participants had to choose options from a closed list, but an “other reasons” option was included in the list.

### 2.4. Variables

The dependent variable was the dichotomous variable “vaccination reason”, which was categorized as health-related or non-health-related and was constructed from the information provided in set 7, “reasons that have led the people to recent vaccination”. For participants that had chosen the option “other reasons”, two researchers analyzed individual answers and classified them as health-related or non-health-related. Discrepancies were solved by consensus. All the other variables were used as independent variables for all the analyses.

### 2.5. Statistical Analysis

The collected data were coded and entered into an Excel (Microsoft Office Excel 2010) database. Data were analyzed using Statistical Package Stata 17. Data were described by frequency distribution and percentages for qualitative variables and by mean and standard deviation (SD) for quantitative variables. Descriptive analysis was conducted for all variables from the whole sample and according to sex. To compare the frequency distributions of data, the Chi-square test was used. To compare means of data, the Student’s *t* test and U Mann–Whitney tests were used. A statistical significance level of *p* < 0.05 had previously been established for all analyzed data. Next, a bivariate analysis of all the variables was carried out based on the variable ‘vaccination reason’, which was a dichotomous variable obtained by grouping all the possible vaccination reasons into two categories: health-related reasons and non-health-related reasons. The health-related reasons were medical recommendation, fear of COVID-19 and fear of transmitting COVID to others. The non-health-related reasons were obtaining the COVID-19 vaccination certificate, being able to travel and work-related reasons. For the answers coded as ‘others’, we classified each individual reason as health-related or non-health-related. Multivariate analysis (logistic regression model) was further used. Regression analysis was conducted using ‘vaccination reason’ as the dependent variable. The independent variables were the selected socio-demographic variables (age, sex, marital status, occupation, children, cohabitation with vulnerable people), compliance with barrier gesture ‘greetings without contact’, self-perceived risk score, vaccine security perception score and FCV 19-S Score. 

## 3. Results

### 3.1. Participants Characteristics

During the study period, 1768 people were vaccinated against COVID-19 in the Region of Murcia: 1054 men (61.07%) and 672 women (38.93%); no information was available on the gender of 42 people (Figure 1). According to nationality, 1255 were non-native to Spain (70.98%) and 513 were Spanish (29.02%) (Figure 2). From the total population of 1768, 798 people were contacted, 196 of which were excluded due to the language barrier, meaning there were 556 people who met the inclusion criteria. A final number of 338 people fully completed the survey. This means a survey participation rate of 60.79% of the population contacted (Figure 3). 

### 3.2. Sample Descriptive Analysis

Most participants lived in the Region of Murcia (97.04%). A total of 188 participants had non-native origins (55.62%), the most frequent being North Africa (43.46%), South and Central America (29.84%) and Eastern Europe (9.42%), with uneven distribution by gender. The average age was 36 years old. Working single men with no education and unemployed married women with secondary-level studies were the most common profiles among participants. Moreover, 11.24% of participants had received vaccination against the flu in the previous campaign. All these socio-demographic data are available in the Appendix A.

In relation to COVID-19 variables (Appendix A), 41.72% of participants had already been infected by SARS-CoV-2, mostly with mild forms of infection (no hospitalization required, no second infections reported). A total of 18.64% of interviewed people were living with a vulnerable person, and 71.30% of them had at least one relative who had suffered from COVID-19; 11.86% of these relatives who suffered the infection had died from SARS-CoV2. Furthermore, 66.86% had followed recommendations about barrier gestures (greetings without contact) in the previous months.

Self-risk perception was similar among men and women (17.25), as were the results in the vaccine security dimension (20.68). The Fear of COVID-19 Scale scores showed differences between men and women, with a significantly higher level of fear among women (11.57 versus 9.967, *p* = 0.0044). All these results are shown in Appendix A.

The reasons for not being vaccinated earlier are shown in Table 1. The main reason for getting the COVID-19 vaccine for most participants was travel restrictions (27.22%), followed by other reasons (20.12%), fear of COVID-19 (13.02%) and medical recommendation (11.83%). Moreover, 78.40% affirmed that the need for the COVID-19 vaccination certificate in restaurants and bars had not been a reason to get the vaccine. Finally, 57% of interviewed people reported non-health-related reasons for getting the vaccine (Table 1).

### 3.3. Bivariate Analysis by Reason for Vaccination (Health-Related Versus Non-Health-Related Reasons)

We performed a bivariate analysis according to the reason for vaccination (health-related versus non-health-related), shown in Table 2. Among participants that received late COVID-19 primary vaccination for health-related reasons, there were more women than men (56.20% women vs. 31.54% men, *p* < 0.001), the mean age was higher (38.22 years old vs. 32.96, *p* = 0.0495), there were more Spanish people than non-Spanish people (54.72% vs. 34.76%, *p* = 0.002) and more participants had Spanish parents (54.37% vs. 35.33%, *p* = 0.002). We found an association between occupation and vaccination for health-related reasons. However, more retired and unemployed participants have been vaccinated for health-related reasons than active workers (55.74% of unemployed and 73.33% of retired participants vs. 34.13% of active workers, *p* = 0.002). We also found an association between marital status and vaccination for health-related reasons, with more married people getting vaccinated for sanitary reasons than single people (52% of married vs. 33.60 of single, *p* = 0.005). Participants with children reported more vaccination for health-related reasons than those who did not have children (53.13% vs. 30.77%, *p* = 0.003).

Regarding COVID-19-related variables (Appendix A), the only one which showed association with vaccination for health-related reasons was compliance with barrier gestures “greetings without contact”: participants who had not followed recommendations about barrier gestures were vaccinated for non-health-related reasons more than people who had followed these recommendations (68.89% versus 51.67%, *p* = 0.007). Participants or their relatives having previous SARS-CoV-2 infections did not show any association with vaccination for health-related reasons, regardless of the severity of the infection.

Concerning the self-perceived risk dimension, participants who became vaccinated for health-related reasons showed higher scores than participants who decided to receive the vaccine for non-health-related reasons (20.35 versus 15.34, *p* < 0.001). In the vaccine security dimension, participants who reported health-related reasons for vaccination scored higher than those who reported non-health-related reasons (22.90 versus 19.54, *p* < 0.001). Similarly, in FCV 19-S, participants reporting health-related reasons for vaccination obtained higher scores than participants reporting non-health-related reasons (13.12 versus 9.30, *p* < 0.001). All data are given in Table 3.

When analyzing reasons for not getting the primary vaccination earlier (Table 4), there was a great number of people reporting non-health reasons for getting vaccinated among participants who expressed the beliefs that vaccines did not work, that the vaccines had been developed too fast or that COVID-19 was not such a threat compared to participants who refused these statements (81.58% vs. 53.45%, *p* = 0.001; 85.71% vs. 55.02%, *p* = 0.006; 85.71% vs. 54.13%, *p* = 0.001, respectively). Among those who reported medical advice against vaccination, there were more participants who finally got the vaccine for health-related reasons than participants who got the vaccine for non-health-related reasons (77.27% vs. 39.52%, *p* = 0.001).

### 3.4. Regression Model: Vaccination for Health-Related Reasons

The regression model (Table 5) showed a significant positive association between vaccination for health-related reasons and female gender (β = 0.72, *p* = 0.044), being married (β = 1.00, *p* = 0.031), cohabiting with a vulnerable person (β = 0.97, *p* = 0.030), compliance with barrier gesture, i.e., “greetings without contact” (β = 0.71, *p* = 0.038) and higher results in the self-risk perception dimension (β = 0.13, *p* = 0.001), vaccine security dimension (β = 0.14, *p* = 0.008) and fear of COVID-19 scale (β = 0.10). The model revealed a negative association between vaccination for sanitary reasons and being an active worker (β = −1.34). The R-squared value for this model was 0.322.

## 4. Discussion

This work presents the originality of studying the characteristics of a late-vaccinated population in the COVID-19 vaccination campaign. Since the entire population had the opportunity to be vaccinated starting in summer 2021, the sample studied consists of people vaccinated between February and May 2022, more than 6 months after the entire population had the opportunity to access to vaccination.

From the total of 1768 people who received primary vaccination during the study period, a total of 556 were both contacted and met the inclusion criteria in the study, of whom 338 (60.79%) agreed to participate. The final participants presented an under-representation of the male and foreign-origin population with respect to that of the total vaccinated in the study, which may have been due to the language barrier. 

The study sample is younger than the average population in the Region of Murcia (90th percentile is 57 years in our population versus 72 years in the Region of Murcia’s population [22]), with a very similar gender distribution (52.98% men in our sample versus 50% in the Region of Murcia’s population). Especially noteworthy is the over-representation of migrant origin (56.21% in our sample versus 14.6% in the population as a whole). This could imply a lower awareness of the younger population with regard to vaccination, an issue described in several works [23,24]. However, the population’s place of birth and belonging to minorities have been associated differently depending on the study and may be influenced both due to lower awareness and access to vaccination [25,26]. Nevertheless, in our sample, we have almost no second-generation Spanish people, which is probably because immigration is a recent phenomenon in Spain and Murcia.

Regarding other characteristics of the sample, it can be seen that cohabiting children had received systematic vaccinations in a proportion similar to that of the general population [27]. Although generally vulnerable cohabitants with these people were vaccinated against COVID-19, their coverage rate was lower than that of the general population at the time the survey was performed (73.9 versus 93.5%) (14). The lower proportion of those vaccinated against influenza in the last campaign can be explained by the lower average age of the sample compared to the general population. An individual or a person close to them having previously suffered from the disease, as well as the severity of that infection, has been associated with a decrease in vaccine reluctance; however, one’s own suffering from COVID-19 reduces vaccination reluctance more than a person close to them being infected [25,28]. In line with this, in our sample, 11.4% had required hospitalization due to COVID-19 and 5.7% had been admitted to the ICU, while 20.7% and 7.95% had a close contact who had been hospitalized for COVID-19 or had required ICU admission.

The main reasons for not vaccinating previously were fear of adverse effects of the vaccine (16.56%), believing that vaccines do not work (14.2%) or not considering that COVID-19 is a threat (9.47%), according to most published works [25,28,29]. However, other reasons which had previously been published in the literature include “not believing in the government”, which is a reason given by only 2.07% of our sample.

Regarding the main reasons for getting vaccinated, the most prevalent were travel reasons (27.2%), fear of contracting COVID-19 (13.02%), medical recommendation (11.83%) and obtaining a vaccination certificate (10.65%). Globally, non-health-related reasons represented 57.41% of all reasons for getting vaccinated. The obligation, the request for a vaccination certificate or the travel requirement have been widely described as reasons that have led to vaccination. For example, a study carried out in the US estimated that 11% of people were vaccinated for these reasons in the 8 weeks after the announcement of the vaccination certificate’s implementation [30]. Similar data have been observed in Belgium [23].

Despite our results, this work has several limitations:

(1) The survey was carried out in a specific population vaccinated at the end of the vaccination process, but we do not have data from the general population. This prevents us from making comparisons with respect to the general population. Likewise, we do not have data from the unvaccinated population, so this study can only present results from the population that, although vaccine-hesitant, finally received vaccination.

(2) The response rate of those contacted is 42.35%, which is somewhat lower than that of other works cited above. As our data show, among the population with late primo-vaccination, there were more foreigners than in the general population. This may be due to the language barrier that prevented 196 people from completing the survey. Moreover, we were not able to contact 970 people. A larger sample size would have allowed us to obtain more statistically robust data. However, we believe that the defined profiles are clear and would not have been modified with a larger sample size.

(3) The survey was conducted at a specific time, crosswise, which limits the knowledge we had about the possible evolution of reticence in the surveyed population.

Despite these limitations, we believe that the data provided can be very useful to address vaccine reluctance against COVID-19 because our work allowed us to classify the reasons for getting vaccinated as being health-related versus non-health related, and the two different profiles were defined. In the bivariate analysis, a relationship was observed between getting vaccinated for health reasons and gender (female), nationality (Spanish origin), marital status (married), living with children, occupation (higher in retired and unemployed people), compliance with non-pharmacological measures (greetings without contact), greater self-perceived risk for COVID-19, higher score on the FCV-19S and greater confidence in the safety of the vaccines. In a complementary way, the factors associated with vaccination for non-health related reasons were gender (male), non-Spanish origin, active workers, marital status (single), not living with vulnerable people, not following the recommendations on barrier gestures, lower self-perceived risk, lower score regarding vaccine safety and on the FCV-19S and greater distrust in the efficacy of the vaccines.

In a systematic review of the literature [28], we found many surveys on vaccine reluctance; however, our study describes the characteristics that lead to vaccination in initially hesitant people who have delayed their vaccination. In this sense, we have only found one publication comparable to our work, a survey carried out in Belgium [24] in which the reasons that led to vaccination in the reticent Belgian population were studied. In that study, the main reasons for getting vaccinated were to facilitate travel (444 [48%]), to recover freedom in day-to-day life (399 [44%]) and due to reported social pressure (387 [42%]), while only 10% would be vaccinated for reasons related to personal protection against COVID-19. This 10% contrasts with the 42.59% of people who were vaccinated for health-related reasons in our work.

For the reasons stated above, this work can be useful for health authorities in relation to the design of communication strategies encouraging the population to be vaccinated and in defining different profiles on which to act. Vaccine hesitancy is an attitude or sentiment, whereas vaccination is an action which is measured to determine vaccine coverage. The period of hesitancy and indecision is a time of vulnerability and opportunity [31]. Vaccine acceptance exists on a continuum, ranging from a minority who stridently oppose all vaccinations to the majority who are willing to accept all recommended vaccines [32]. In our work, two parts of this continuum are described. The first one is those with health-related reasons for vaccination, who are apparently easier to address with the correct information when provided in a timely manner. The second one is those with non-health-related vaccination hesitancy reasons; these people present greater resistance and may sometimes require additional measures or incentives to become vaccinated. We must not, however, forget that people may periodically update their beliefs and intentions, and it is important to provide support throughout the decision process [33]. For the reasons stated above, this work can be useful for health authorities as knowing the different COVID-19 vaccination hesitancy profiles on which to act will aid in the design of specific communication strategies for the population to be vaccinated.

## 5. Conclusions

Spain is one of the countries with the highest vaccination coverage in the EU. Our work clearly defines two profiles among the people who were finally vaccinated, despite previous hesitancy. In terms of addressing the profile of those vaccinated for health reasons, correctly carried out information campaigns can play a large role in achieving their vaccination. However, we have also described a profile of people vaccinated for non-health-related reasons. Information campaigns may be insufficient and other incentives may be needed to compel this group to undergo vaccination.

## Figures and Tables

**Figure 1 vaccines-11-00732-f001:**
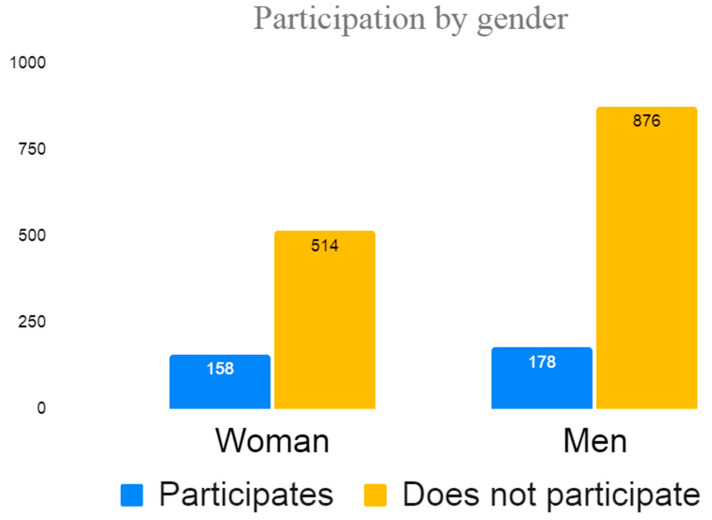
Participation distribution by gender.

**Figure 2 vaccines-11-00732-f002:**
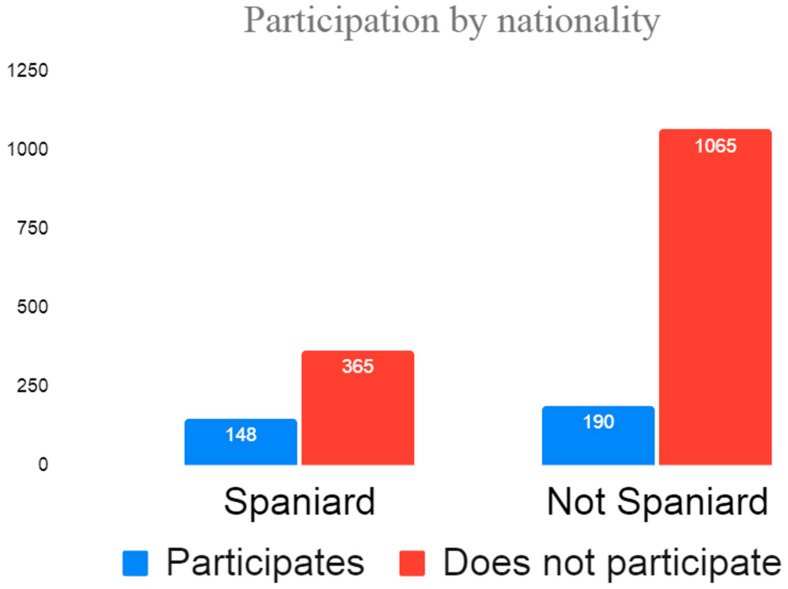
Participation distribution by nationality.

**Figure 3 vaccines-11-00732-f003:**
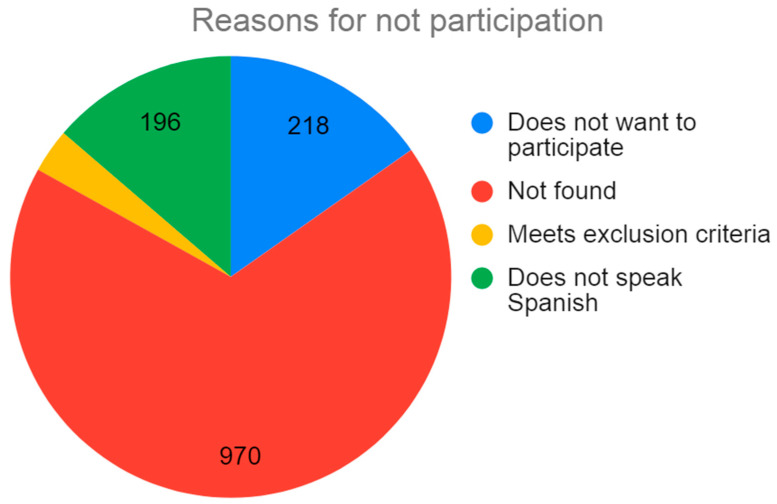
Distribution of non-participation reasons.

**Table 1 vaccines-11-00732-t001:** Reason for not being vaccinated earlier (part 6 of the survey) and for getting the COVID-19 vaccine (part 7 of the survey).

	Variable	*n* (%) Mean (SD)	*n* (%) Mean (SD)		*p* Chi2/StudentT/Mann–Whitney
		Total Sample: 338	Men	Women	
Reasons for not being vaccinated earlier	Reason for not getting vaccine: adverse events				0.242
	No	282 (83.43%)	155 (54.96)	127 (45.04)	
	Yes	56 (16.56%)	26 (46.46)	30 (53.57)	
	Reason for not getting vaccine: I don’t think vaccines work				0.18
	No	290 (85.80%)	151 (52.07)	139 (47.93)	
	Yes	48 (14.20%)	30 (62.50)	18 (37.50)	
	Reason for not getting vaccine: doctor told me not to				0.073
	No	311 (92.01%)	171 (54.98)	140 (45.02)	
	Yes	27 (7.99%)	10 (37.04)	17 (62.96)	
	Reason for not getting vaccine: I prefer to wait				0.955
	No	310 (91.72%)	148 (53.62)	128 (46.38)	
	Yes	28 (8.28%)	33 (53.23)	29 (47.77)	
	Reason for not getting vaccine: COVID-19 vaccines have developed too fast				0.236
	No	310 (91.72%)	169 (54.52)	141 (45.48)	
	Yes	28 (8.28%)	12 (42.86)	16 (57.14)	
	Reason for not getting vaccine: I do not believe in government				0.085
	No	331 (97.93%)	175 (52.87)	156 (47.13)	
	Yes	7 (2.07%)	6 (85.71)	1 (14.29)	
	Reason for not getting vaccine: COVID-19 is not such a threat				0.15
	No	306 (90.53%)	160 (52.29)	146 (47.71)	
	Yes	32 (9.47%)	21 (65.63)	11 (34.38)	
	Reason for not getting vaccine: I am afraid of needles				0.411
	No	324 (95.86%)	172 (53.09)	152 (46.91)	
	Yes	14 (4.14%)	9 (64.29)	5 (35.71)	
	Reason for not getting vaccine: I cannot because of another treatment				0.13
	No	328 (97.04%)	178 (54.27)	150 (45.73)	
	Yes	10 (2.96%)	3 (30.00)	7 (70.00)	
	Reason for not getting vaccine: others				0.89
	No	208 (61.54%)	112 (53.85)	96 (46.15)	
	Yes	130 (38.46%)	69 (53.08)	61 (46.92)	
	The need for a COVID-19 certificate in bars and restaurants has been a reason for getting the vaccine				0.004 *
	No	265 (78.40%)	131 (49.43)	134 (50.57)	
	Yes	73 (21.60%)	50 (68.49)	23 (31.51)	
	Have you received a full vaccination schedule?				0.597
	No	122 (36.09%)	63 (51.64)	59 (48.36)	
	Yes	216 (63.91%)	118 (54.63)	98 (45.37)	
	Reasons for not getting full vaccination schedule				0.782
	One shot is enough	10 (7.75%)	6 (60.00)	4 (40.00)	
	Adverse events myself	6 (4.65%)	02 (33.33)	4 (66.67)	
	Close person with adverse events	4 (3.10%)	2 (50.00)	2 (50.00)	
	Others	109 (84.50%)	53 (48.62)	56 (51.38)	
	Reasons to get the vaccine				0.001 *
	Get vaccination certificate	36 (10.65%)	24 (66.67)	12 (33.33)	
	Travel	92 (27.22%)	59 (64.13)	33 (35.87)	
	Work-related reasons	27 (7.99%)	19 (70.37)	8 (29.63)	
	Medical recommendation	40 (11.83%)	12 (30.00)	28 (70.00)	
	Fear of getting COVID-19	44 (13.02%)	19 (43.18)	25 (56.82)	
	Fear of transmitting COVID-19	31 (9.17%)	16 (51.61)	15 (48.39)	
	Others	68 (20.12%)	32 (47.06)	36 (52.94)	
	Reasons to get the vaccine				<0.001 *
	Non-health-related	155 (57.41%)	102 (65.81)	53 (34.19)	
	Health-related	115 (42.59%)	47 (40.87)	68 (59.13)	

SD: standard deviation; * *p* < 0.05.

**Table 2 vaccines-11-00732-t002:** Bivariate analysis by reason for vaccination (socio-demographic variables).

	Variable	*n* (%) Mean (SD)		*p* Chi2/ StudentT/Mann–Whitney	Col percentage (Adjusted Residual)	
		Non-Health-Related	Health-Related		Non-Health-Related	Health-Related
Socio-demographic variables	Gender			<0.001 *		
	Man	102 (68.46)	47 (31.54)			
	Woman	53 (43.80)	68 (56.20)			
	Autonomous Community			0.194		
	Murcia	150 (56.82)	114 (43.18)			
	Others	5 (83.33)	1 (16.67)			
	Age	33.96 (11.09)	38.22 (15.57)	0.0495 *		
	Under 18			0.002 *		
	No	155 (58.94)	108 (41.06)			
	Yes	0 (0.00)	7 (100.00)			
	Spanish			0.001 *		
	No	107 (65.24)	54 (34.76)			
	Yes	48 (45.28)	58 (54.72)			
	Spanish parents			0.002 *		
	No	108 (64.67)	59 (35.33)			
	Yes	47 (45.63)	56 (54.37)			
	Parents’ birth country			0.012 *		
	Occidental Europe	4 (100.00)	0 (0.00)		100.00 (1.502)	0.00 (−1.502)
	Eastern Europe	5 (38.46)	8 (61.54)		38.46 (−2.042) *	61.54 (2.042) *
	North Africa	57 (76.00)	18 (24.00)		76.00 (2.791) *	24.00 (−2.791) *
	Central and southern Africa	10 (76.92)	3 (23.08)		76.92 (0.975)	23.08 (−0.975)
	Central and South America	27 (51.92)	25 (48.08)		51.92 (−2.277) *	48.08 (2.277) *
	Asia	2 (40.00)	3 (60.00)		40.00 (−1.162)	60.00 (1.162)
	Others	4 (57.14)	3 (42.86)		57.14 (−0.415)	42.86 (0.415)
	Education			0.638		
	No studies	27 (65.85)	14 (34.15)			
	Primary studies	31 (57.41)	23 (42.59)			
	Secondary studies	63 (55.76)	48 (43.24)			
	Higher education	34 (53.13)	30 (46.88)			
	Occupation			0.002 *		
	Active	110 (65.87)	57 (34.13)		65.87 (3.580) *	34.13 (−3580) *
	Retired	4 (26.67)	11 (73.33)		26.67 (−2.478) *	73.33 (2.478) *
	Student	14 (51.85)	13 (48.15)		51.85 (−0.615)	48.15 (0.615)
	Unemployed	27 (44.26)	34 (55.74)		44.26 (−2.360) *	55.74 (2.360) *
	Marital status			0.005 *		
	Single	83 (66.40)	42 (33.60)		66.40 (2.775) *	33.60 (−2.775) *
	Married	59 (47.97)	64 (52.03)		47.97 (−2.869) *	52.03 (2.869) *
	Divorced	12 (70.59)	5 (29.41)		70.59 (1.135)	29.41 (−1.135)
	Widower	1 (20.00)	4 (80.00)		20.00 (−1.707)	80.00 (1.707)
	Cohabiting with vulnerable person			0.068		
	No	131 (60.09)	87 (39.91)			
	Yes	24 (46.15)	28 (53.85)			
	Cohabiting vulnerable person vaccinated			0.001 *		
	No	11 (91.67)	1 (8.33)			
	Yes	16 (37.21)	27 (62.79)			
	Children			0.003 *		
	No	86 (66.67)	43 (33.33)			
	Yes	69 (48.94)	72 (51.06)			
	Children insurance-covered vaccines			0.124		
	No	9 (69.23)	4 (30.77)			
	Yes	60 (46.88)	68 (53.13)			
	Children non-insurance-covered vaccines			0.544		
	No	43 (50.59)	42 (49.41)			
	Yes	24 (45.28)	29 (54.72)			
	Flu vaccine in 21–22 campaign			0.402		
	No	141 (58.26)	101 (41.74)			
	Yes	14 (40.00)	14 (50.00)			
	Flu vaccine in previous campaign			0.563		
	No	127 (58.26)	91 (41.74)			
	Yes	28 (53.85)	24 (46.15)			

SD: standard deviation; * *p* < 0.05.

**Table 3 vaccines-11-00732-t003:** Bivariate analysis by reason for vaccination (self-perceived risk, vaccine security, FCV 19-S).

	Variable	*n* (%) Mean (SD)		*p* Chi2/StudentT/Mann–Whitney
		Non-Health-Related	Health-Related	
Self-perceived risk	COVID-19 vaccine makes me feel safer at work	2.92 (1.38)	3.80 (1.38)	<0.001 *
	COVID-19 vaccine helps me protect my family	3.14 (1.42)	4.24 (0.99)	<0.001 *
	COVID-19 vaccine makes me feel comfortable with my family	3.01 (1.40)	4.16 (0.96)	<0.001 *
	COVID-19 vaccine will decrease my risk of getting COVID-19	3.01 (1.36)	3.97 (1.17)	<0.001 *
	COVID-19 vaccine will decrease the risk of hospitalization	3.27 (1.26)	4.19 (0.94)	<0.001 *
	Self-perceived risk total	15.34 (5.84)	20.35 (4.37)	<0.001 *
Vaccine security	Approved COVID-19 vaccines are safe	2.92 (1.13)	3.88 (0.86)	<0.001 *
	COVID-19 vaccines adverse events are similar to other vaccines	3.17 (1.16)	3.58 (1.04)	0.0027 *
	Vaccine high-speed development has decreased safety	3.39 (1.20)	3.19 (1.23)	0.1909
	I trust Drug Agencies’ work	3.26 (1.10)	3.81 (0.80)	<0.001 *
	I trust Health Authorities’ recommendations	3.57 (1.21)	4.38 (0.82)	<0.001 *
	COVID-19 vaccine benefits are greater than risks	3.23 (1.21)	4.04 (0.87)	<0.001 *
	Vaccine security total	19.54 (4.48)	22.90 (3.10)	<0.001 *
FCV 19-S	I am afraid of COVID-19	1.67 (1.01)	2.72 (1.41)	<0.001 *
	I feel uncomfortable when thinking about COVID-19	1.47 (0.86)	2.24 (1.27)	<0.001 *
	My hands sweat when I think about COVID-19	1.18 (0.59)	1.38 (0.92)	0.0936
	I am afraid of dying from COVID-19	1.35 (0.78)	2.15 (1.43)	<0.001 *
	I get nervous when listening to news about COVID-19	1.44 (0.91)	1.95 (1.24)	<0.001 *
	I cannot sleep because of COVID-19	1.11 (0.45)	1.30 (0.62)	0.0009 *
	My heart accelerates when I think I can get COVID-19	1.09 (0.37)	1.38 (0.88)	0.0006 *
	FCV 19-S total	9.30 (3.66)	13.12 (6.19)	<0.001 *

FCV 19-S: Fear of COVID-19 Scale; SD: standard deviation; * *p* < 0.05.

**Table 4 vaccines-11-00732-t004:** Bivariate analysis by reasons for not getting primary vaccination earlier.

	Variable	*n*% Mean (SD)		*p*/ Chi2/StudentT/Mann–Whitney
		Non-Health-Related	Health-Related	
Reasons for not getting vaccinated earlier	Reason for not getting vaccine: adverse events			0.08
	No	135 (59.73)	91 (40.27)	
	Yes	20 (45.45)	24 (54.55)	
	Reason for not getting vaccine: I do not think vaccines work			0.001 *
	No	124 (53.45)	108 (46.55)	
	Yes	31 (81.58)	7 (18.42)	
	Reason for not getting vaccine: medical advice			0.001 *
	No	150 (60.48)	98 (39.52)	
	Yes	5 (22.73)	17 (77.27)	
	Reason for not getting vaccine: I prefer to wait			0.512
	No	126 (56.50)	97 (43.50)	
	Yes	29 (61.70)	18 (38.30)	
	Reason for not getting vaccine: COVID-19 vaccines have developed too fast			0.006 *
	No	137 (55.02)	112 (44.98)	
	Yes	18 (85.71)	3 (14.29)	
	Reason for not getting vaccine: I do not believe in government			0.052
	No	150 (56.60)	115 (43.40)	
	Yes	5 (100.00)	0 (0.00)	
	Reason for not getting vaccine: COVID-19 is not such a threat			0.001 *
	No	131 (54.13)	111 (45.87)	
	Yes	24 (85.71)	4 (14.29)	
	Reason for not getting vaccine: I am afraid of needles			0.629
	No	150 (57.69)	110 (42.31)	
	Yes	5 (50.00)	5 (50.00)	
	Reason for not getting vaccine: I cannot because of another treatment			0.248
	No	152 (58.02)	110 (41.98)	
	Yes	3 (37.50)	5 (62.50)	
	Reason for not getting vaccine: others			0.12
	No	102 (61.08)	65 (38.92)	
	Yes	53 (51.46)	50 (48.54)	
	The need for a COVID-19 certificate in bars and restaurants has been a reason for getting the vaccine			<0.001 *
	No	105 (50.00)	105 (50.00)	
	Yes	50 (83.33)	10 (16.67)	
	Have you received a full vaccination schedule?			0.977
	No	55 (57.29)	41 (42.71)	
	Yes	100 (57.47)	74 (42.53)	
	Reasons for not getting full vaccination schedule			0.129
	1 shot is enough	3 (100.00)	0 (0.00)	
	Adverse events myself	2 (40.00)	3 (60.00)	
	Close person with adverse events	4 (100.00)	0 (0.00)	
	Others	52 (57.78)	41 (40.20)	

SD: standard deviation; * *p* < 0.05.

**Table 5 vaccines-11-00732-t005:** Regression model: vaccination for health-related reasons.

Variable	β	*p*	CI 95%	
Age	0.02	0.356	−0.02	0.05
Gender: woman	0.72	0.044 *	0.02	1.42
Occupation: student				
Active worker	−1.34	0.021 *	−2.48	−0.2
Retired	−0.57	0.628	−2.86	1.72
Unemployed	−0.87	0.172	−2.12	0.38
Marital status: single				
Married	1,00	0.031 *	0.09	1.91
Divorced	0.47	0.575	−1.16	2.09
Widower	2.63	0.088	−0.39	5.66
Cohabiting with a vulnerable person: yes	0.97	0.030 *	0.09	1.84
Children: yes	−0.12	0.791	−1.04	0.79
Compliance with “greetings without contact”: yes	0.71	0.038 *	0.04	1.39
Self-perceived risk total score	0.13	0.001 *	0.05	1.21
Vaccine security dimension total score	0.14	0.008 *	0.04	0.26
Fear of COVID-19 scale total score	0.1	<0.001 *	0.03	0.16

CI: confidence interval; * *p* < 0.05.

## Data Availability

The data presented in this study are available on request from the corresponding author.

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
