# Peer review of "Profiles of People Who Carried Out Late Primary Vaccination against COVID-19 in the Region of Murcia"

_vaccines, 2023, doi:10.3390/vaccines11040732_

Round 1

Reviewer 1 Report

The authors have made an interesting attempt at “Analysis of the reasons that have led people late vaccinated to beat their COVID-19 vaccination hesitancy.” The manuscript is interesting; however, the authors need to justify the scientific writing manuscript. Some of the general comments are provided below:

1.     The research gap must clearly be stated and keeping in line with the same, the research goals for the paper are to be presented.

2.     The exact technical contribution and novelty of the work need to be stated explicitly.

3.     Did the authors conduct a pilot study to validate the clarity and comprehensibility of survey questions, if yes, what is the test score of Cronbach's alpha?

4.     The authors have not mentioned the type of vaccine these participants took and what kind of vaccines were available at that time in Spain. In case, there was more than one vaccine, what was the preference of people and why?

5.     Interpretation and analysis of experimental observations are extremely important - authors must pay proper attention to this in the manuscripts.

6.     The figures should be improved with a large font size of legends and figure size as well.

7.     What is the difference between table 3 and table 7?

8.     The authors should comment on more participation of foreigners as compared to local individuals.

9.     The article could use a little more attention to focus and maintain the argument. The article is very dense and there is almost too much detail. The authors should reduce the tables as it looks repletion of information in different tables.

10. Please check some grammatical errors like line 123.

Author Response

  1. The research gap must clearly be stated and keeping in line with the same, the research goals for the paper are to be presented. We have better described the research gap in the introduction.
  2. The exact technical contribution and novelty of the work need to be stated explicitly. We have better explained the novelty of the work and contribution in the discussion.
  3. Did the authors conduct a pilot study to validate the clarity and comprehensibility of survey questions, if yes, what is the test score of Cronbach's alpha? Yes. Fear of COVID-19 Scale is a scale validated in Spanish (bibliographical reference 21 in the article). For the full questionnaire, a pilot study was performed in the general population and those questions with comprehension problems were redrafted, but alpha Cronbach’s score wasn’t calculated. 
  4. The authors have not mentioned the type of vaccine these participants took and what kind of vaccines were available at that time in Spain. In case, there was more than one vaccine, what was the preference of people and why? We have added in the introduction that COVID-19 Vaccination Strategy in Spain did not contemplate the possibility for citizens to choose the type of vaccine and that they were given the vaccine available for their age group in the vaccination centre of their area. Nevertheless, in order to give more information we have also added that, in 2022, the only vaccines administered in the Region of Murcia were mRNA vaccines.
  5. Interpretation and analysis of experimental observations are extremely important - authors must pay proper attention to this in the manuscripts.
  6. The figures should be improved with a large font size of legends and figure size as well: We have added figures in the results section, participants characteristics, but we also add them here as well as the link to the editable images:

Graphs

  1. What is the difference between table 3 and table 7? Table 3 is a descriptive analysis by sex, while table 7 is by reason for vaccination. 
  2. The authors should comment on more participation of foreigners as compared to local individuals. In the sample there is an overrepresentation of migrant origin (56.21% in our sample versus 14.6% in the population as a whole). As it is explained in the article, this could imply a lower awareness of the foreign population or and accessibility to vaccination. Nevertheless, in our sample we don’t practically have second generation Spaniards, probably this is because immigration is a recent phenomenon in Spain and Murcia.
  3. The article could use a little more attention to focus and maintain the argument. The article is very dense and there is almost too much detail. The authors should reduce the tables as it looks repletion of information in different tables. We have only left 5 tables and the others we have changed them to supplementary material.
  4. Please check some grammatical errors like line 123. Solved after revision by translation service.

Reviewer 2 Report

Dear editor,

Thank you for the kind invitation to review this manuscript. 

Abstract

- Slightly confused about the use of globally in the abstract when it is a local study.

- The direction of the association should be mentioned. 

- The abstract does not correlate well with the study title. 

- There is minimal elaboration about the reasons for people who were initially vaccine hesitant to go for vaccinatio. 

- The conclusion sdoes not tie in with the results highlighted

Introduction

- Some odd sentence phrasing

-> "Public Health vaccination policies in Spain do not imply mandatory vaccines"

-> "being one of the least punished 70 autonomous 59 communities in Spain"

- It would be helpful to cite some international data on high vaccine hesitancy globally and even among healthcare workers at the start to set the context of this manuscript.

to cite the following relevant articles

-> https://www.ncbi.nlm.nih.gov/pmc/articles/PMC8402587/

-> https://www.ncbi.nlm.nih.gov/pmc/articles/PMC8402587/

- It will be helpful to set the context by citing literature for changes in behaviour for initially vaccine hesitant people and their associated reasons in other countries and Spain (if available)

- The evidence gap that the authors aim to bridge should be highlighted in greater details

Methods

- Use the STROBE checklist for the study

- How was the questionnaire for this study designed?

- Were any attempts made to pilot the questionnaire?

- It will be helpful to describe any vaccination campaigns in Spain during the study period and the general policy in Spain about COVID-19 vaccination

-> e.g. what are the vaccination programs implemented?

Results

- Suggest to avoid over-focusing on the inclusion of patients

-> To report characteristics of included patients only in the first paragraph.

- All abbreviations in the tables should be defined.

- There are excessive number of tables

- Suggest to transfer some to the supplementary materials or shorten current tables

- In the Table 9

-> clarification regards "active", and vaccine security needs to be made regarding what they refer to

Discussion

- What are the implications of the study findings and what can be done to improve patient's willingness to undergo vaccination?

- Were there any findings that differ from initial hypothesis? 

Author Response

Abstract

- Slightly confused about the use of globally in the abstract when it is a local study. We have removed “globally” from the abstract. 

- The direction of the association should be mentioned: We have added in the abstract as well as in the results that the association is positive.

- The abstract does not correlate well with the study title. We have modified that abstract and the title for better comprehension and correlation.

- There is minimal elaboration about the reasons for people who were initially vaccine hesitant to go for vaccination. We have added the most reported reasons, both for health-related and non-health-related ones.

- The conclusion does not tie in with the results highlighted. We have modified it.

Introduction

- Some odd sentence phrasing. Modified after revision by translation service.

-> "Public Health vaccination policies in Spain do not imply mandatory vaccines" 

-> "being one of the least punished 70 autonomous 59 communities in Spain". 

- It would be helpful to cite some international data on high vaccine hesitancy globally and even among healthcare workers at the start to set the context of this manuscript and to cite the following relevant articles: https://www.ncbi.nlm.nih.gov/pmc/articles/PMC8402587/ We have added both suggestions.

- It will be helpful to set the context by citing literature for changes in behaviour for initially vaccine hesitant people and their associated reasons in other countries and Spain (if available). For our knowledge, the only available work that has studied a change in behaviour for initially COVID-19 vaccine hesitant people is the Belgian one (bibliographical reference 24 in the article). The originality of our work lies in the fact that there are no other works that have investigated this change in behaviour in previously hesitant people.

- The evidence gap that the authors aim to bridge should be highlighted in greater details. We have added a sentence at the end of the introduction. 

Methods

- Use the STROBE checklist for the study. We used the STROBE checklist, but we have now added a paragraph to correctly describe variables before the statistical analysis section. We have also modified some discussion paragraphs in order to better follow the STROBE checklist.

- How was the questionnaire for this study designed? We have better explained it in this section. 

- Were any attempts made to pilot the questionnaire? Yes. Fear of COVID-19 Scale is a scale validated in Spanish (bibliographical reference 21 in the article). For the full questionnaire, a pilot study was performed in the general population and those questions with comprehension problems were redrafted, but alpha Cronbach’s score wasn’t calculated. 

- It will be helpful to describe any vaccination campaigns in Spain during the study period and the general policy in Spain about COVID-19 vaccination. -> e.g. what are the vaccination programs implemented?  We have completed the introduction indicating that COVID-19 Vaccination Strategy is the same for the whole country and a description of it. We also have explained that February is usually, in the Region of Murcia, the last month of the flu vaccination campaign for people aged 60 years or older and those with risk conditions, with a low number of daily doses administered, in which, having availability of doses, those people not belonging to risk groups who so wish can be vaccinated. The final of this vaccination campaign coincided with the start of our study. 

Results

- Suggest to avoid over-focusing on the inclusion of patients -> To report characteristics of included patients only in the first paragraph. We have summarised them and changed the figure’s order.

- All abbreviations in the tables should be defined. We have added CI meaning in the Abbreviations section and all the abbreviations in each table after each one.

- There are an excessive number of tables. Suggest to transfer some to the supplementary materials or shorten current tables. We have left only 5 tables and the others we have changed them to supplementary material.   

- In the Table 9 -> clarification regards "active", and vaccine security needs to be made regarding what they refer to. We have clarified everything on the table. 

Discussion

- What are the implications of the study findings and what can be done to improve patient’s  willingness to undergo vaccination? We have modified discussion and conclusions in order to explain it better.

- Were there any findings that differ from the initial hypothesis? There is no “initial hypothesis” because it is a descriptive study. We tried to characterise and define profiles, so we did not assume anything.

Round 2

Reviewer 1 Report

The authors have modified the manuscript and now it is acceptable for the publication. 

Reviewer 2 Report

The changes made by authors are acceptable